# Advanced Monitoring of Manufacturing Process through Video Analytics

**DOI:** 10.3390/s24134239

**Published:** 2024-06-29

**Authors:** Nisar Hakam, Khaled Benfriha, Vincent Meyrueis, Cyril Liotard

**Affiliations:** 1Arts et Métiers, Institute of Technology (AMIT), 75013 Paris, France; khaled.benfriha@ensam.eu (K.B.); vincent.meyrueis@ensam.eu (V.M.); 2ERM Automatismes, 84200 Carpentras, France; c.liotard@erm-automatismes.com

**Keywords:** video analytics, digitization, data deviation, detection, machine, visual, industry 4.0

## Abstract

The digitization of production systems has revolutionized industrial monitoring. Analyzing real-time bottom-up data enables the dynamic monitoring of industrial processes. Data are collected in various types, like video frames and time signals. This article focuses on leveraging images from a vision system to monitor the manufacturing process on a computer numerical control (CNC) lathe machine. We propose a method for designing and integrating these video modules on the edge of a production line. This approach detects the presence of raw parts, measures process parameters, assesses tool status, and checks roughness in real time using image processing techniques. The efficiency is evaluated by checking the deployment, the accuracy, the responsiveness, and the limitations. Finally, a perspective is offered to use the metadata off the edge in a more complex artificial-intelligence (AI) method for predictive maintenance.

## 1. Introduction

The framework of Industry 4.0 focuses on deploying three computing approaches: edge, fog, and cloud. These structures enable the integration of image analysis studies on the shop floor, as seen in Figure 1. Edge computing includes all modules, servers, or desktops situated on the operation technology (OT) network. Fog computing allows the usage of the information technology (IT) network, offering better performance due to high-performance equipment. Cloud computing is considered an open configuration platform, providing fast response and preconfigured processing blocks.

### 1.1. Industrial Data

Data mining is fundamental in modern industrial sectors, aiming to optimize and adjust operations, as explained by Tambare et al. (2022) [1]. However, essential challenges such as system integration, communication, and collaboration must be tackled. The goal is to maintain a continuous production cycle, preventing unexpected machine breakdowns. This objective aligns with smart supervision and edge computing, where data are contextualized into performance indicators, expanding system monitoring.

Several studies have explored the potential of operational images to extract metadata from isolated systems. Various image-based methods allow the measurement of operational and environmental data from system operations. However, image processing requires parameter tuning and stable lighting, as presented by Mirbod et al. (2022) [2]. Simultaneously, AI approaches facilitate metadata prediction if a sufficient, unbiased dataset is available, as illustrated by Caggiano et al. (2019) [3].

### 1.2. Video Analytics in Edge Computing

Considering the objective of Industry 4.0, Bazhenov et Korzun (2019) discusses the application of event-driven video services for monitoring in edge-centric Internet of Things (IoT) environments [4]. It emphasizes that video analytics in IoT environments should not rely on transferring raw video data to a central processing center (like cloud-based systems). Instead, it proposes a model where multiple edge video capture devices process data streams locally for event detection. These events can be simple occurrences within a single video stream or complex events derived by correlating events across multiple streams and incorporating contextual information from the IoT environment. The authors highlight the importance of microservice architecture, event-driven models, and context-based mechanisms in constructing these video services. They also present a demo implementation of a video service for personnel recognition around production equipment, showcasing its use in digital monitoring for smart manufacturing processes.

Simultaneously, Kim et al. (2019) present a novel monitoring approach for small and medium-sized enterprises (SMEs) called KEM (keep an eye on your machine) [5]. The system uses a low-cost vision device like a webcam and open-source technologies to collect and process operational data from computer numerical control (CNC) machine tools. A prototype was tested with a 3-axis CNC milling machine, and the results showed that the system was easy to integrate and could be applied to legacy machine tools without significant changes or cost barriers. The authors believe that this approach can change monitoring operations, reduce operational time and energy consumption, and lessen the environmental impact of sustainable production in SMEs.

These two applications present initial roadmaps for video analytics. Bazhenov et Korzun (2019) [4] suggest that their event-driven model, which is a key element of their overall concept, can be extended to other applications beyond manufacturing, such as object recognition and recognition of interaction between people and physical objects. On the other hand, Kim et al. (2019) [5] consider the application of their approach with other manufacturing equipment, such as turning centers, parts feeders, and 3D printers.

### 1.3. Scope

This paper aims to reinforce the models of Bazhenov et al. (2019) and Kim et al. (2019) [4,5]. This reinforcement consists of optimizing the processing power for the edge module by preventing data latency, machine reconfiguration, and security risks. Section 2 highlights some video analytics in manufacturing, considering the limitations, the structural implementation, and the accuracy of the algorithm. The approach is tested on a CNC lathe machine by contextualizing video frames into operational data.

## 2. Image Processing and AI Vision

Image processing is based on filters and pixel calculations, while AI vision is a modern analysis technique that learns pixel deviations and correlates unseen images with pre-trained ones. Both techniques are considered artificial intelligence (AI) methods. Presently, they are used in manufacturing processes to contextualize video data, such as operative parameters and process faults.

An investigation of 11 research studies shows the importance of video analytics in manufacturing processes according to five criteria, as seen in Table 1.

### 2.1. Additive and Subtractive Manufacturing

The study by Mazumder (2015) proposed a design for a metal additive manufacturing machine [6]. They incorporated a video sensor for constant defect monitoring. While suitable for zero-point design, the approach did not discuss advanced image analysis, achieving less than 80% efficiency due to limitations in integrating reliable measurements using edge computing.

Feature point localization is also an image processing technique. Nuchitprasitchai et al. (2017) used the RANSAC and SIFT algorithms to identify key points between a 3D-printed part and its virtual representation [8]. This method controlled printing accuracy and ensured continuous operation without faults, but it slowed down printing by a factor of four.

Augmented reality (AR) is another technology for assessing printing faults. Ceruti et al. (2017) implemented an innovative approach using AR technology [9]. They projected a simulation from the CAD file onto the printing bed, detecting 1-mm deviations and running filament defects between the projected and real product. However, integration failed when not enough feature points were recognized between real and simulated products using the SURF algorithm.

In addition to classical recognition, AI algorithms predict data. Delli et Chang (2018) trained an SVM, a machine learning method trained to detect running filaments and fault prints during the process [10]. They installed a camera with an adapted graphic card, achieving 90% accuracy and less than 1 s response time, but reproducing the work required, including print pauses and high-performance servers.

Caggiano et al. (2019) developed an approach to transform images from a selective laser-melting (SLM) printer into operational data [3]. Points, such as the applicability of other machines and overfitting probability, were not discussed. Simultaneously, training and deployment required computational resources, making it only applicable to an integrated fog or cloud network.

Along with additive manufacturing, Wang et al. (2022) used existing methodologies to implement a robot welding process [14]. They utilized deep learning methods, like CNN and DCNN, to measure operational parameters: welding speed, current, and torch angle. Surface measurement for the welded area was necessary to calculate process parameters using the DCNN, requiring fog or cloud computing for optimized training and deployment.

AM presents only one type of creation of new products from 3D CAD files. In industry, these manufacturing machines are not always deployed due to uncertainty in final production. Thus, a discussion about SM is fundamental to understanding video analytics’ applicability in the field.

Video analysis on SM machines focuses on a detection model for the machining context, such as parameters, clipping angle, etc. A deeper investigation is crucial to understand the requirements for integration with real production data. Currently, AI vision approaches are not often deployed due to biased datasets. For example, Giusti et al. (2020) trained a CNN model to predict the surface roughness (Ra) of a manufactured product [12]. The model was trained on publicly available datasets like W300, ASP, STAVAX, and V720, achieving 98% accuracy. However, the authors noted that Ra and binary quality classification are insufficient for quality control. Future work aims to extend this model to predict more measurements about product surface quality.

With low computational hardware, it is possible to manipulate a manufacturing machine, as demonstrated by Halawi Ghoson et al. (2023) [15]. They used an embedded vision sensor on a robot to manipulate a 3-axis milling machine. The robot initiated a production cycle with only the smart-mounted camera, using image processing approaches. Unlike Wang et al. (2022) [14], this application had limitations due to lighting effects and machine security.

### 2.2. Dataset Creation

Training AI methods require non-biased datasets. Cai et al. (2021) proposed a generative noisy image model [13]. Their objective was to train a model that adds random noise to an image to augment the dataset, making it more suitable for training. Simultaneously, an operator cannot distinguish if the generated image presents an environmental issue or a process issue. Hence, it is necessary to develop an online edge computing device capable of extracting relevant parameters to create datasets. These datasets might serve as references to enhance AI models since they are based on real images and contextual data.

### 2.3. Edge Detection

Edge detection is considered an old-fashioned recognition method. It depends on filters like Sobel, Prewitt, etc. The common approach is the Canny edge detector. Abid Hasan et Ko (2016) proposed a method for segmentation using 3D measurement technologies and the Canny edge detector [7]. Images are passed through erosion and dilation filters to remove noise before being processed by the Canny detector. The detector measures edges inside the image, creating a depth map that segments the image into sections corresponding to the tool, part, and machine interior. This technique is under evolution due to false predictions from unwanted random noises, addressed in a detailed study by Kalbasi et Nikmehr (2020) [11]. They developed an adaptable Canny edge detector, increasing contour detection even with noisy images. These results suggest researching a technique using the Canny detector instead of more sophisticated and complex SIFT, SURF, and AI algorithms.

In conclusion, video analytics have the potential to integrate smart manufacturing. According to the studies, three conditions are required: effective edge computing modules, performance devices, and stable lighting. A general expansion method is proposed to allow machine data extraction using classical recognition algorithms. This method is then tested on an SM machine, presenting critical aspects for data measurement in a production line.

## 3. Proposed Methodology

The review explores the conditions for building an effective image data contextualization method. One requirement is data measurement using edge computing modules.

The data of a non-connected CNC machine are enclosed within the control loop, requiring a generic method to measure essential characteristics. A methodology is proposed to extract relevant information. It is a 4-stage phase, starting with the specific needs, refer to Figure 2.

### 3.1. Specific Needs

Initially, it is essential to specify the data to identify. This objective depends on each system, as they differ in shape, software, and operational data. Based on these needs, a vision system is investigated. For example, one camera is not sufficient for 3D printer supervision, so two cameras are required to supervise the printed product. The vision module also assesses tool quality or raw part presence.

An optimal place is chosen to mount the camera without affecting the process or data collection. The camera’s view angle must prioritize capturing the process. As a result, the machine presents several constraints requiring technical implementations.

### 3.2. Constraints

Machine constraints play a role in finding the operational data. These constraints are examined to prevent false installations that might affect the supervision device. It is beneficial to correlate between the needs and the constraints. For example, vibration is an important parameter to monitor during manufacturing. As a result, the camera is placed in front of the process to detect the chatter of the workpiece. Additionally, noisy images are prevented by mounting the module on a rigid body. Finally, cable management is also crucial to prevent interference with the production process.

### 3.3. Camera Specifications

The aim is to ensure data measurement using only a camera and edge detection. Thus, a suitable camera is essential for monitoring a machine and communicating the metadata to the supervision level. Industrial vision systems are often not accessible due to budget or technical support constraints.

The design of a custom vision sensor should fit industry standards. For example, Costa (2018) has identified the Raspberry Pi with its camera as a customizable system [16]. Additionally, this vision module satisfies working-area constraints, according to Kumar et al. (2018) [17]. The module is sized 65 × 56 × 12.5 mm, making it possible to add support or fix it to the walls of a machine without perturbing the process.

Moreover, the literature review points out that lighting has a major effect on images. Thus, it is important to overexpose the scene, ensuring good image quality with no noise [8].

A final condition is the capturing rate. Using the image acquisition ratio (IAR), it is possible to understand the camera’s capability to capture subsections of a process. IAR represents the proportionality between the camera’s sampling frequency and the machine’s working speed (refer to Equation (1)).
(1)$$IAR=\frac{FPS_{camera}(frame\;per\;second)}{Speed_{machine}(rotation\;per\;second)}$$

### 3.4. Video Analytics

The algorithmic configuration varies between low-level analysis and AI vision. The focus is on low-level analysis, integrating the concepts of edge detection.

For a manufacturing machine, several properties are required. Data contextualization is either on the edge or on remote servers. This approach introduces edge integration. Edge computing is then configured on the vision module using a low computational algorithm, like the Canny detector.

The Canny edge detector is a multi-stage algorithm that combines several techniques to detect edges in images:Noise reduction with Gaussian Blur

A Gaussian filter is applied to smooth the image and reduce noise. The filter kernel is defined by the Gaussian function
(2)$$G\left(x,y\right)=\left(\frac{1}{2\pi \sigma^{2}\;}\right)*e^{\frac{-{(x}^{2}+y^{2})}{2\sigma^{2}}}$$
where sigma controls the degree of smoothing. A larger sigma results in more blurring.

2.Gradient calculation using Sobel filters

Sobel filters are applied to calculate the intensity gradient of the image. Two filters are used, one for the horizontal gradient ($G_{x}$) and one for the vertical gradient ($G_{y}$). The gradient magnitude (G) and direction (θ) are then calculated:(3)$$G=\sqrt{\left(G_{x}^{2}+G_{y}^{2}\right)}\;$$
(4)$$\theta =arctan\left(\frac{G_{y}}{G_{x}}\right)$$

3.Non-maxima suppression

Thinning of edges by suppressing non-maximum gradient magnitudes. Only pixels where the gradient magnitude is a local maximum in the direction of the gradient are kept.

4.Double thresholding

Two thresholds are used—a high threshold (${thresh}_{high}$) and a low threshold (${thresh}_{low}$).

Strong edges (G above ${thresh}_{high}$) are marked as definite edges. Weak edges (C between ${thresh}_{low}$ and ${thresh}_{high}$) are potential edges.

5.Edge tracking by hysteresis

Weak edges are connected to strong edges to form continuous contours. Weak edges not connected to strong edges are discarded.

Binary images reveal a strong ratio for object detection and recognition. Additionally, it serves as a refined image for the Canny detector, refer to Figure 3. In the end, these contextual data are relatively small packets, making it advantageous to use the MQTT protocol for transmission.

Some calculations are presented in the next section since these techniques are based on the needs and the parameters to monitor. It is important to validate that the obtained data responds to the identified needs.

The deployment focuses on the CNC lathe machine, which generates noisy images due to environmental disturbances. These machines are commonly deployed in manufacturing lines because they produce products faster.

## 4. Experimentation

The experimentation is conducted on a pedagogic platform to develop technological bricks for the Industry 4.0 framework. The focus is on the CNC lathe machine of this platform.

### 4.1. Objectives and Constraints

The methodology is tested on a lathe machine. It treats the case of how to properly integrate the vision system.

First, this machine is an automated unit designed to operate by an operator without the ability to make external connections. The data are enclosed inside the internal closed control loop. This concept applies to many machines used throughout the industrial manufacturing lines. Thus, it is relevant to have external monitoring units, like a vision system, to track the process and ensure its integration within the Industry 4.0 framework. The metadata aids in scheduling maintenance operations, minimizing machine breakdown, and enhancing supervision. Therefore, the vision module must:Measure the process parametersAssess the quality of the toolCheck the roughnessDetect part presence

These needs are common and require an adapted analytical approach. To proceed with sensor configuration, machine constraints are exploited in Table 2.

The edge module is placed inside the machine to guarantee a continuous view of the process. It also prevents missing out on important scenes due to the turret displacement. Figure 4 resumes the general conception where the essential mechanical systems, like the spindle speed and the turret, are presented.

### 4.2. Camera Selection and Validation

A 2-dimensional analysis with one camera is sufficient for a lathe machine. The camera selection, introduced in Section 3.4, defines the sampling frequency and the ability to process the image. The ratio is found by looking at the highest machining parameters of the lathe. In our case, it is 2500 rotations per minute (RPM). For an adequate sampling frequency, IAR should be greater than 1.1, allowing the capturing of an image every rotation (refer to Equation (2)).
(5)$$IAR=\frac{FPS_{camera}}{Speed_{machine}}$$
(6)$$IAR=1.1=\frac{FPS_{camera}}{2500\left(rpm\right)*\frac{1}{60}}=\frac{FPS_{camera}}{41.66(rotation/second)}$$
(7)$$FPS_{camera}=1.1*41.66=45.825\;fps$$

As a result, the selected camera should be rated at least 46 fps. According to Costa (2018), the pi-camera of Raspberry Pi achieves a sampling rate of 60 fps.

After considering the constraints of Table 2 and selecting the Raspberry Pi 4 with its camera module, the installation is realized, as shown in Figure 5.

This implementation achieves a sampling frequency of 60 Hz with a control processing unit (CPU) usage of 40%. This means that the potential is not fully used, and the board can handle some video analytics. Simultaneously, the operating system, Raspbian, is fully customizable and adaptable for multi-threading with its 4 cores and 4 Gigabytes (GB) of random-access memory (RAM).

At this stage, the initial steps of the approach are well integrated for the supervision of a lathe machine. The next phase is to define the data treatment flow, allowing the investigation of the process parameters.

### 4.3. Video Analytics

According to the objective in Section 4.1, a data flow is created to visualize and ensure valid data extraction (refer to Figure 6).

The expected outputs of the detection sequence are summarized in Table 3. These outputs are transferred to the supervisory control level in text format to facilitate the operator’s interpretations.

#### 4.3.1. Object Presence

Detecting objects in the chuck allows verification if the automatic feeding system has performed its task correctly or if the workpiece has been ejected during the production process. This detection permits the control unit to stop the machine, preventing unwanted time loss and potential malfunctions.

The described studies are based on CNN for object detection. However, they require high-performance servers to turn efficiently, which is not available in most manufacturing machines. A less computational approach is to use the black-to-white ratio (BWR), defined as follows:(8)$$BWR=\frac{black\;pixels}{white\;pixels}$$

This ratio is only applicable to binary images. The original image is transformed into grayscale and then converted to a binary image. This approach efficiently detects material presence with a total computational time of 10 milliseconds per frame (refer to Figure 7).

If the BWR is lower than 0.80, it means a product is detected inside the chuck. Otherwise, no material is detected. Correlating this result with a start operation button enhances machine intelligence by alerting any material conditions and object ejection during the process.

Simultaneously, the transformation from pixel to centimeter is given using the theorem of Thales, explained in Figure 8.

This diagram shows the detection of lengths L1 and L2; see Figure 7. Using the theorem of Thales, it is given:(9)$$\frac{f}{D}=\frac{l}{L}$$
$$L=\frac{l*D}{f}=l*\frac{30}{3.04*10^{-1}}$$

To obtain L in cm, it is crucial to know the camera specifications about the pixel size. It allows the transformation of the length measured in pixels into micrometers. In this case, it is around 1.12 µm.

#### 4.3.2. Clamping

Clamping issues affect the final product quality. Therefore, detecting any misalignment of the raw part before manufacturing is advantageous. Clamping is detected using the Hough transform and geometric formulas. The Hough transform involves five steps successive steps:Step 1: edge detection using the optimized Canny detector algorithm.Step 2: extract the line parameters from the edge image by creating a voting function.Step 3: select the thresholds to consider true lines and omit false positives.Step 4: apply the threshold to the extracted lines and obtain the true positives.Step 5: draw the line on the image and save it locally.

The presence of a part indicates that the horizontal surface of the raw material receives the most votes as a line. Simultaneously, it is crucial to know the object’s presence to avoid algorithm misinterpretation.

Applying the transform to the video stream, five lines are detected: three horizontals and two verticals (refer to Figure 9). The vertical lines showcase a well-placed part, while the horizontal lines highlight a 3° deviation. The algorithm sends a warning to the operator to consider adjustments.

Additional experiments are conducted to detect the tolerance, which is around ±0.14° on average. The angle order is validated to prevent misinterpretation. The measurement is deployed at two thresholds, 50 and 80, for accurate identification. The total response time is found to be 2 milliseconds.

Based on the lighting effects, it is possible to tune the thresholds of the Hough transform. In this use case, we overexposed the scene. Then, the algorithm is parameterized by setting the threshold of votes, where a line is defined as a true line.

#### 4.3.3. Tool Control

Tool supervision is the capability to predict whether a tool performs machining well. In AI vision, the Siamese Neural Network (SNN) compares a reference tool state and the current tool state to verify tool wear. However, it is possible to perform edge detection combined with the Hough transform to find the tool angle and deduce the tool wear. For example, the tools of a lathe machine have specific geometric reference profiles. If the tool sustains damage, the angle will vary drastically; see Figure 10.

For the detection of the tool angle, it is essential to validate the output angle order. Another experiment focuses on the detection of the tool and detecting the angle. The measured angle is found to be always the last one predicted by the essential algorithm. The detection time is measured around 1.5 milliseconds.

#### 4.3.4. Cutting Parameters

Measuring machine parameters is crucial since the machine does not allow external communication. The lathe machine has three parameters: spindle rotation, cutting speed, and depth of cut.

A blue landmark is placed to detect the tool. The vision module detects the blue color based on specified masks. The cutting speed is measured by detecting the tool displacement in the image. For example, at time T1, the tool is at position P2; at time T2, the tool moved to position P2′ (refer to Figure 11).

The cutting speed is found:(10)$$cutting\;speed=\frac{distance}{time}\;in\;PPS\;\left(\frac{pixels}{second}\right)$$
where
(11)$$distance=\sqrt{{\left(x_{P2^{\prime }}-x_{P2}\right)}^{2}+{\left(y_{P2^{\prime }}-y_{P2}\right)}^{2}}\;in\;pixels$$
time=T2−T1 in seconds

The unit does not conform to the physical value. Based on the camera configuration, mapping allows the discovery of the speed of rotation per minute.

The machine’s rotational speed is measured using the chuck’s diameter and the time for one rotation. With an IAR greater than 1, one image is captured per turn. A red landmark is placed on it to detect the spindle. The vision module measures the rotation by timing the red circle’s exact position in the image.

Finally, the depth of cut is found by evaluating the pixel difference between two consecutive horizontal passes. In this case, it is found to be 2 pixels equivalent to 1 mm.

#### 4.3.5. Quality Control

Quality control is a critical evaluation that presents a wide study area to automate the process. Giusti et al. (2020) used CNN models to find the roughness [12]. However, deploying it on an edge device reduces performance drastically. The Hough transform method detects lines in an image, representing deviations from the final surface (refer to Figure 12).

The detection sequence is divided into idle and production states. Figure 7, Figure 8, Figure 9, Figure 10, Figure 11 and Figure 12 summarize the detection approaches during these phases. The data measured are contextual and require a total time of 100 milliseconds per image. The following section investigates the importance of the approach relative to other studies.

## 5. Discussion

The proposed method augments data communication for unconnected CNC machines, evaluating camera integration limitations. Additionally, a suitable edge algorithm responds to the defined needs.

Our setup detects the chuck holding a partially machined product with an incorrect clamping angle. The machine tool is then examined, and a notice is sent to the machinist about its lifecycle. This preprocess analysis serves as initialization before manufacturing. During the process, the camera measures the cutting parameters, validating correct execution. Finally, the system assesses product quality, detecting possible surface imperfections and estimating roughness based on expert opinion.

The proposed approach responds to the five identified needs for machine supervision. The algorithm has a total timespan of 100 milliseconds, ensuring real-time response and data contextualization. Compared to other approaches like SIFT, SURF, RANSAC, and CNN, our model performs better on an edge device with limited resources. Table 4 summarizes the comparison between literature methods and our approach.

The properties are obtained by reimplementing their methods on our machine with the Raspberry Pi of CPU @1.5 GHz and 4 GB RAM. The accuracy for feature point detection is low, with an execution time near 1 s. With AI vision, the controller heated up, presenting an execution time of 2 s. Our approach is suitable for the edge setup and is known to have low computational power.

## 6. Limitations

The algorithm can fail to assess machine parameters without scene overexposure. Additionally, some algorithm glitches occur during angle measurements and quality assessment. These deviations are avoided by considering all environmental constraints, such as lighting, shadows, noise, and offsets.

Another major limitation occurs when the manufacturing machine’s shape changes. Since the camera extracts features from a 2D process considering the symmetrical shape of products, for a 3-axis milling machine, we require updating the 5-stage algorithm to fit geometrical specifications.

The transition toward Industry 4.0 is limited by the lack of data availability from old manufacturing machines. Our approach proves the concept of data extraction from such a system, helping industries achieve a smoother transition with less financial burden.

## 7. Conclusions

The paper aimed to define an approach to improve the integration of video analytics in manufacturing. We were able to implement a 4-stage method that allows the digitization of any non-connected machine. This scope is defined by the Reference Architecture Model Interface 4.0 (RAMI 4.0). Thus, it was possible to transfer essential data to the supervisory control level of a production line. These data serve to remotely monitor machines and automate critical scenarios.

The implementation focuses on precisely defining the objective to prevent over-calculation. In addition, it is optimized to use the full potential of a low-cost edge computing device. In short, the Raspberry Pi turned with a total computational power of 75% and near real-time processing with 100 milliseconds per image. The standards of the industry are also considered, so the MQTT protocol is applied for data transfer.

In conclusion, the external smart vision system upgrades closed control systems like CNC machines. We explained the deployment of a vision system, extracting metadata. In our use case, the lathe machine is now equipped with an advanced monitoring device describing object presence, clamping, tool state, operational parameters, and surface roughness. With the wireless edge device, the machine becomes interoperable, transforming it into a CPS. This approach allows a gradual evolution toward a digitalized version with adequate communication and data processing.

## 8. Perspectives

Using the data extracted from a CNC machine, a data pool is created in real time. This data pool serves as a reference to train more advanced approaches, like CNN and DCNN, which help to reorchestrate the manufacturing process. These approaches require fog and cloud computing. Simultaneously, more complex AI models can be trained to refine the quality control procedure.

## Figures and Tables

**Figure 1 sensors-24-04239-f001:**
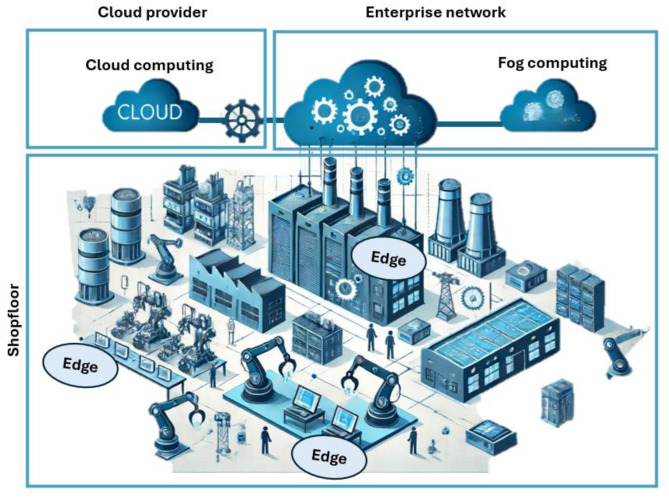
Computing architectures inside the framework of Industry 4.0.

**Figure 2 sensors-24-04239-f002:**
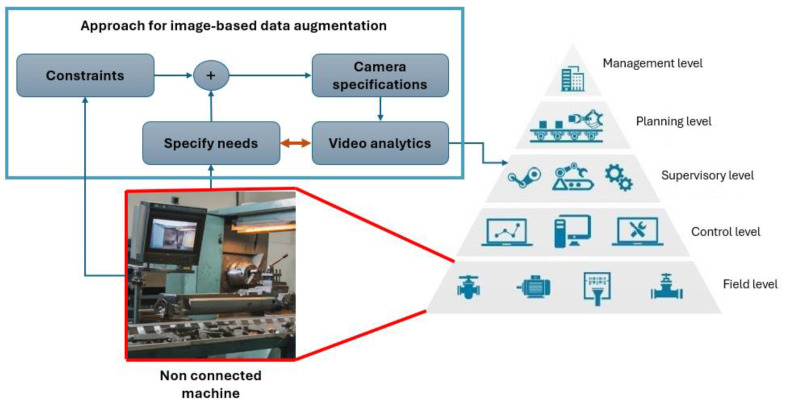
Proposed methodology for an edge computing module.

**Figure 3 sensors-24-04239-f003:**
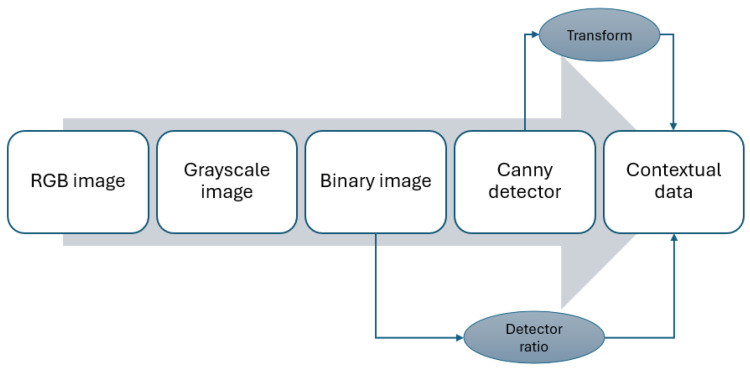
Image flow to extract contextual data.

**Figure 4 sensors-24-04239-f004:**
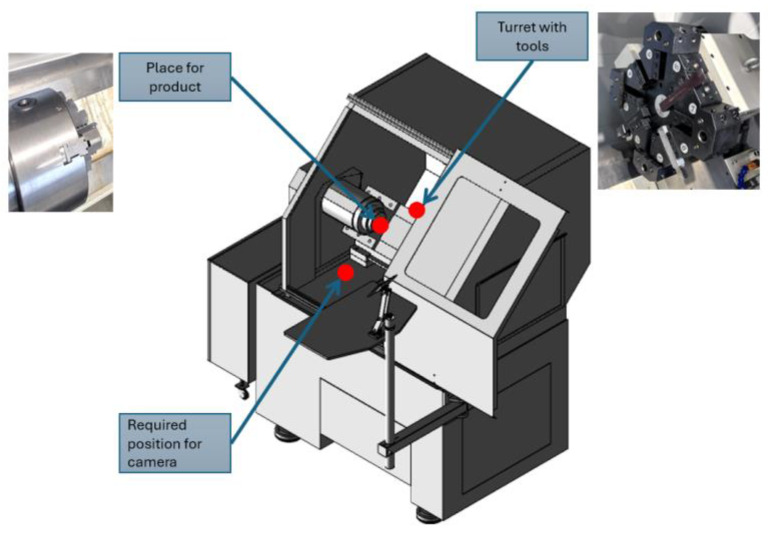
Geometrical constraints of CNC lathe machine from TORMACH, Madison, WI, USA.

**Figure 5 sensors-24-04239-f005:**
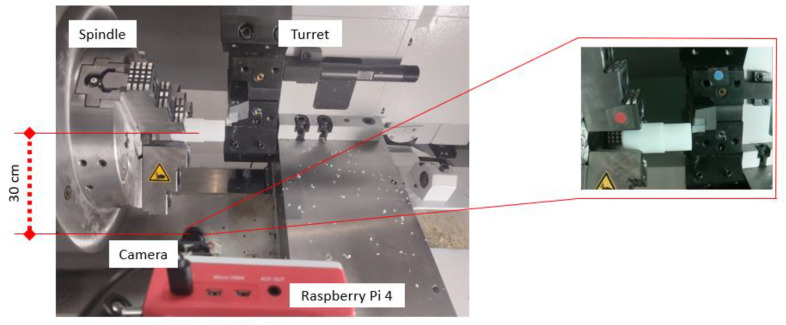
Hardware integration on the lathe machine.

**Figure 6 sensors-24-04239-f006:**
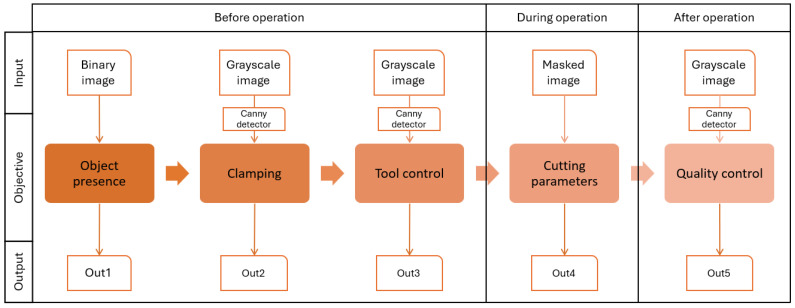
Objectives of video analytics for the lathe machine.

**Figure 7 sensors-24-04239-f007:**
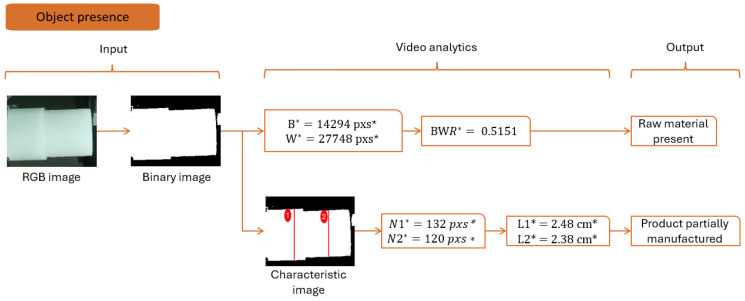
Analytics to detect object presence and manufactured parts. * B—black pixels; W—white pixels; BWR—black-to-white ratio; N1—number of pixels for segment 1; N2—number of pixels for segment 2; L1—length of segment 1; L2—length of segment 2; pxs—abbreviation for pixels; cm—abbreviation for centimeters.

**Figure 8 sensors-24-04239-f008:**
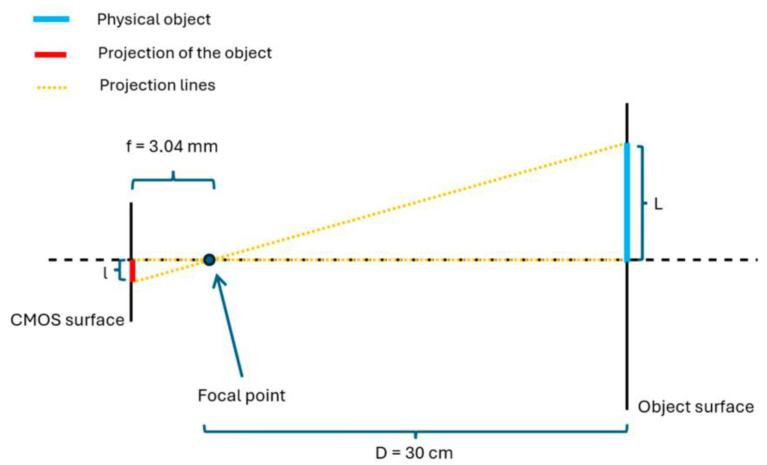
Length proportionality between real object and image.

**Figure 9 sensors-24-04239-f009:**
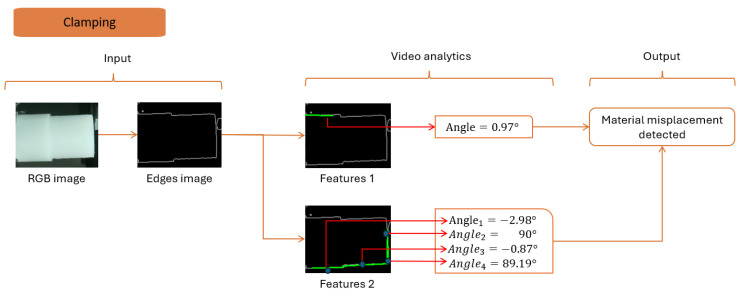
Analytics to part position and orientation.

**Figure 10 sensors-24-04239-f010:**
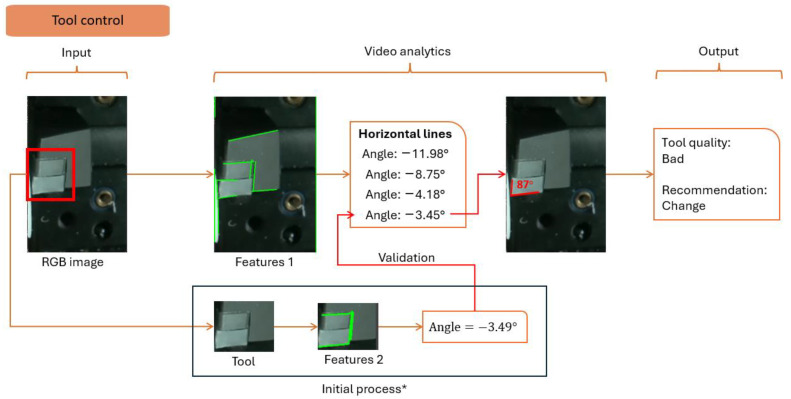
Analytical approach to measure the tool angle. * Initial process is only launched once to validate the position of angles extracted from Features 1.

**Figure 11 sensors-24-04239-f011:**
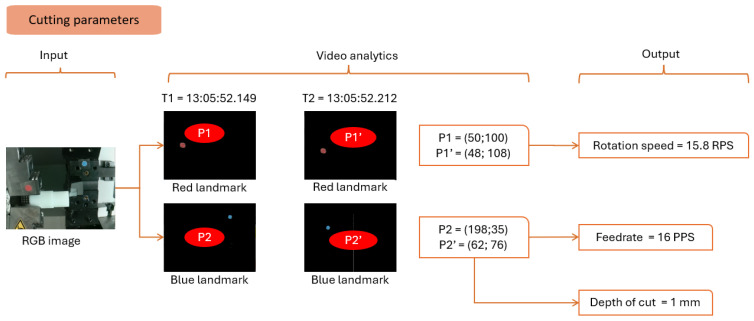
Analytical approach to measure the operational parameters.

**Figure 12 sensors-24-04239-f012:**
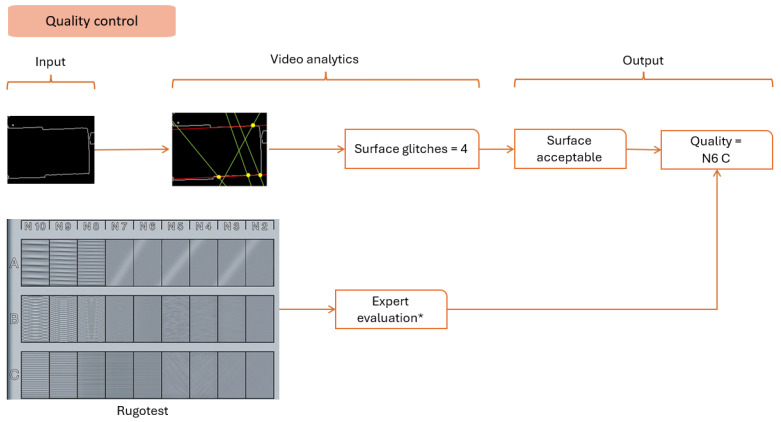
Analytical approach to measuring the quality of the surface. * Expert evaluation is still done manually since the detection of stripes is being researched as a future extension.

**Table 1 sensors-24-04239-t001:** A comparative review of image-based studies.

Study	Process	Algorithm	Network	Accuracy	Gap
Mazumder (2015) [6]	AM *	SVM * PCA *	Edge/Fog	<80%	Nonsufficient
Abid Hasan et Ko (2016) [7]	Detection	Canny	Edge	>95%	Applicability in manufacturing
Nuchitprasitchai et al. (2017) [8]	AM *	SIFT * RANSAC *	Edge	>90%	Code modification
Ceruti et al. (2017) [9]	AM *	SURF *	Edge	>90%	Refine error
Delli et Chang (2018) [10]	AM *	SVM *	Edge/Fog	90%	Add checkpoints
Caggiano et al. (2019) [3]	AM *	DCNN *	Fog/Cloud	99.4%	Single process
Kalbasi et Nikmehr (2020) [11]	Detection	Canny	Edge	--	Applicability in manufacturing
Giusti et al. (2020) [12]	SM *	CNN *	Fog/Cloud	98%	Apply in situ automatic control
Cai et al. (2021) [13]	Detection	PNGAN *	Fog/Cloud	>95%	Applicability in manufacturing
Wang et al. (2022) [14]	AM *	CNN * DCNN *	Fog/Cloud	98%	Include more capabilities
Halawi Ghoson et al. (2023) [15]	SM *	Mask	Edge	99%	Nonadopted for online monitor

* AM—additive manufacturing; SM—subtractive manufacturing; SVM—support vector machine; PCA—principal component analysis; SIFT—scale invariant feature transform; RANSAC—random sample consensus; SURF—speed up robust features; CNN—convolutive neural network; PNGAN—pixel-level noise-aware generative adversarial network; DCNN—deconvolutional neural network.

**Table 2 sensors-24-04239-t002:** Constraints for physical integration of vision system.

Type of Constraint	Description	Resolution
GEOMETRICAL	Machine compact design. Little space for operator manipulation.	Camera facing process. See machine Figure 4. Minimal distance 30 cm.
ENERGY	Input power is 220Vac. Vision modules require 5Vdc with 1.5Adc.	Adapter with a rating of 5Vdc and 2 A
CABLE MANAGEMENT	High magnetic field causing errors in data.	Wi-Fi connection. Only power supply.
AIR PRESSURE	Air pressure to clean the product. Vibration errors on camera.	Use a rigid support to fix the camera module.
ACCESS	Security door closed during manufacturing.	Camera placed inside the machine. Refer to Figure 4.

**Table 3 sensors-24-04239-t003:** Output characterization for video analytics.

Output	Positive Detection	Alternative Detection
OUT 1	Raw part detected	Manufactured part detected
OUT 2	Material right alignment	Misalignment detected
OUT 3	Good tool quality	Bad tool quality
OUT 4	Cutting speed; Depth of cut; Feed rate	No landmarks detected
OUT 5	Acceptable roughness	Too many surface glitches

**Table 4 sensors-24-04239-t004:** A comparison between recent studies and our approach.

Approach	Reference	Hardware	Time	Accuracy
SIFT + RANSAC	[8]	CPU @ 1.5 GHz 4 Gb RAM	900 ms	81%
CNN	[9]	CPU @ 1.5 GHz 4 Gb RAM	2 s	98%
SURF	[10]	CPU @ 1.5 GHz 4 Gb RAM	800 ms	82%
**Our approach**	--	CPU @ 1.5 GHz 4 Gb RAM	100 ms	95%

## Data Availability

All codes and configurations are made available upon request for confidentiality reasons.

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
