# Peer review of "Advanced Monitoring of Manufacturing Process through Video Analytics"

_sensors, 2024, doi:10.3390/s24134239_

Round 1
Reviewer 1 Report
Comments and Suggestions for Authors
Summary
This work focuses on leveraging images from a camera system and its technological environment to monitor the manufacturing process on a CNC lathe. There are major comments (described below), should be considered.
Remarks to the Author: Please see the full comments.
1- It is recommended to highlight the key contributions of the proposed work over other existing works in the same field. And the authors need to be more specific in addressing the problems that are examined in this work and when the system performs well?
2- In general, the introduction section is poorly organized and needs to be rearranged.
3- Table 1 is important but it is weak and should have more comparative criteria among existing works in this field. Besides, more recent related work needs to be added with pointing out the used techniques, results and gaps.
4-Is the proposed system compared with other existing systems based on accuracy? Can the authors please clarify this point in the manuscript to show the strength of this work?
5- How the roughness of the proposed work has been checked in real time? Please present this point more clearly. It is recommended to make a comparative table with other recent related works to show the robustness of the proposed work.
6-There are many grammatical errors that need to be checked in the whole manuscript.
7- The conclusion in any research paper should contain proposed topics, summarize the main points of the work, include a discussion of its importance, and include some directions of future work. These points need to be addressed in the conclusion section. Besides, it is better to make this section as one paragraph.
Comments on the Quality of English LanguageModerate editing of English language required
Author Response
Dear Reviewer,
Please find attached our point to point responses. We also updated our manuscript to respond to all the presented comments,
Regards

Reviewer 2 Report
Comments and Suggestions for Authors
[summary]
This paper introduces a method for designing and integrating smart video system to detect the presence of raw parts, measure process parameters, assess tool status and check roughness in real time. The method considers both performance and the processing speed.
[strengths]
+ This paper presents some visualisations for understanding this area clearly.
+ The experimental setups are presented in details.
+ The limitations of existing solutions are outlined in the paper, e.g., Sec. 2.
[weaknesses]
Major:
- The paper organisation looks quite strange, especially Sec. 2. The authors should rearrange these sections to make them clear enough to readers.
- It would be better to have some descriptions presented in the table, e.g., Table 1, rather than simply listing methods into different categories.
- The contributions are not described clearly in the Introduction (or sec. 2), it should be improved and clearly outlined so that readers can clearly know what are the major contributions/insights of the paper wants to deliver.
- The approach section is a bit too sparse, and the connections between subsections are weak. The authors should describe their method in an easy-to-understand, well-designed modules with justifications.
- The authors presented figures; however, these figures are not explained properly and clearly in either captions or texts.
- Experimental analysis appears to be quite limited. The authors should provide more experimental results on performance and speed evaluations, with detailed analysis and discussions.
- It would be better to clearly describe some limitations and future work as a separate section.
Minor:
- Eq. E1 in Line 193 appears a bit strange.
- Most figures are in quite low resolutions.
- The tables and figures presented in the paper should be improved to make them look nicer, as a scientific research paper.
Considering all the factors, I would conclude that this paper is not ready at this stage.
Comments on the Quality of English LanguageEnglish usage could be improved.
The organisation of the paper also requires improvement.
Author Response

(The authors gave the same response as above.)

Reviewer 3 Report
Comments and Suggestions for Authors
Comments and Suggestions for Authors
Brief Summary: The manuscript "Advanced monitoring of manufacturing process through video analytics" aims to leverage video analytics for real-time monitoring of a CNC lathe's manufacturing process. The study proposes a methodology for designing and integrating intelligent video modules to monitor various process parameters and assess tool status in real time. The team evaluates the approach for accuracy, responsiveness, and limitations and discusses future improvements.
General Concept Comments:
- Introduction: The introduction provides a general background but needs more depth to present the novelty and comprehensive context of the study. Additionally, some relevant references need to be included or updated. For instance, citing more recent and relevant studies will enrich the discussion around Industry 4.0 and edge computing (Lines 23-40). A more thorough explanation of the specific contributions of this study compared to existing literature is required.
- Research Design: The research design is sound but could benefit from more detailed explanations in some areas. The methodology for integrating the video analytics system into manufacturing needs a detailed explanation. For example, the section describing the identification and installation of hardware (Lines 175-185) needs more clarity on the selection criteria and setup process. Consider adding detailed procedural steps and diagrams to enhance understanding.
- Methods: The methods described need more detail to ensure their reliability and superiority over other options. The writers must convince the reader that their method is the most suitable rather than just functional. The manuscript mentions specific algorithms and techniques for image processing and data extraction but must explain why they chose these over other methods. For example, they need to provide more technical details and visual aids to justify using the Hough transform instead of other advanced methods (Lines 318-323). Including a comparison table that evaluates factors such as execution time, memory size, result accuracy, and detection probability would be beneficial. Alternatively, they could reference a recent article that has already made such comparisons and rely on its recommendations.
- Results: The presentation of results is unclear and lacks coherence. The figures and tables need better integration into the text to ensure the results are easily interpretable. For example, Figure 3 (Line 295) and its components must be more detailed to clearly show how the data is analyzed and the key findings. Improving the clarity and organization of this section is essential.
- Conclusions: The conclusions need more support from the results presented. The authors need to link their claims to the data and analysis provided in the results section. The authors need to link their claims to the data and analysis provided in the results section. For instance, the statement about the methodology's effectiveness in real-time monitoring (Lines 387-389) needs specific examples and quantitative data from the study for support.
While the study addresses an important and relevant topic, the others should significantly improve the presentation and methodological details to meet the standards expected by the journal Sensors. Addressing these issues will greatly enhance the manuscript's clarity, coherence, and scientific impact.
Recommendation: Reconsider after Major Revisions. The manuscript needs substantial improvements in methodology description, results presentation, and language editing.
Comments on the Quality of English Language- Quality of English: Moderate editing of the English language is required. The manuscript contains several grammatical errors and awkward phrasing that detract from the overall readability. For example, the sentence in Lines 156-158 must be clarified and rephrased for better understanding. Consider a thorough language review to improve clarity and professionalism. For instance, for Lines 156-158, Try something like this: "For example, a 3D printer might need more than a simple RGB camera. It also requires a thermal camera 157 to supervise the fuse temperature".
Significant improvements are needed in the language quality to meet the standards expected by the journal Sensors.
Author Response

(The authors gave the same response as above.)

Round 2
Reviewer 1 Report
Comments and Suggestions for Authors
The Title: Advanced monitoring of manufacturing process through video analytics
Some comments were handled properly, others were not. Therefore, the manuscript still needs further modification to be ready for publication. The following comments must be made correctly.
1-However, the general problem statement is clear; the specific problem statement still needs to be clearly defined.
It was mentioned in the cover letter that the goal is clearly written, and the issue was resolved in the updated version in lines 48-54 and highlighted in blue and green, but it was not mentioned clearly. Please check this issue carefully.
It is recommended to add the following statement to the paper “the aim is to provide a way to contextualize data with minimal error and process disruptions)”
In addition, it is preferable to discuss the method used to diagnose errors in more justifications.
2- It was stated that “…. to prevent data latency, machine reconfiguration, and security risks, highlighted in Section 2.”. What do the authors mean by highlighted in section 2?
3- The introduction section is still weak, and in general, more references should be added to the manuscript to make it more robust.
Moreover, the structure of the paper can be added to be more organized.
4-The presentation and quality of the figures still needs more improvement, especially from Figure 7 to the end. Moreover, each figure needs further explanation with a brief definition about the parameters mentioned within it and how some of them are calculated (any mathematical formula needs a reliable source).
5-The conclusion section should be improved again based on the previous comments and generally, avoid the use of in-text references in it.
6- I think that the format of reference 16 needs to be checked.
Comments on the Quality of English Language
Minor editing of English language required
Author Response
Dear Reviewer,
please find our point to point response in the attached file.
Regards,

Reviewer 2 Report
Comments and Suggestions for Authors
After major revision, the overall manuscript looks much better.
The authors should do a minor revision on
(i) equations: please label the equations in the paper,
(ii) figures/tables: please reorganise the figures and tables to make them look nicer.
Comments on the Quality of English LanguageThe authors should do a proofread for the whole paper.
Author Response
Dear Reviewer,
Please find attached our point to point response to your comments
Regards,

Reviewer 3 Report
Comments and Suggestions for Authors
Can be published but minor editing of English language required
Comments on the Quality of English Languageminor editing of English language required
Author Response
Dear Reviewer,
Thank you for taking your time to review our paper. Please find below an explanation about our updates.
Comment 1
Can be published but minor editing of English language required
Response to comment 1:
We proofread the article and done some minor English modifications.